# BGN Secreted by Cancer-Associated Fibroblasts Promotes Esophageal Squamous Cell Carcinoma Progression via Activation of TLR4-Mediated Erk and NF-κB Signaling Pathways

**DOI:** 10.3390/ijms262412024

**Published:** 2025-12-13

**Authors:** Hiroki Yokoo, Yu-ichiro Koma, Naozane Nomura, Rikuya Torigoe, Masaki Omori, Takashi Nakanishi, Shoji Miyako, Takaaki Nakanishi, Takayuki Kodama, Manabu Shigeoka, Yoshihiro Kakeji, Masafumi Horie

**Affiliations:** 1Division of Molecular and Genomic Pathology, Department of Pathology, Kobe University Graduate School of Medicine, Kobe 650-0017, Hyogo, Japan; h450@med.kobe-u.ac.jp (H.Y.); 2033605m@stu.kobe-u.ac.jp (N.N.); rikuarw@med.kobe-u.ac.jp (R.T.); 239m863m@stu.kobe-u.ac.jp (M.O.); 215m857m@stu.kobe-u.ac.jp (T.N.); shoji224@med.kobe-u.ac.jp (S.M.); nthawk@med.kobe-u.ac.jp (T.N.); takodama@med.kobe-u.ac.jp (T.K.); mshige@med.kobe-u.ac.jp (M.S.); mhorie@med.kobe-u.ac.jp (M.H.); 2Division of Gastro-Intestinal Surgery, Department of Surgery, Kobe University Graduate School of Medicine, Kobe 650-0017, Hyogo, Japan; kakeji@med.kobe-u.ac.jp; 3Division of Hepato-Biliary-Pancreatic Surgery, Department of Surgery, Kobe University Graduate School of Medicine, Kobe 650-0017, Hyogo, Japan

**Keywords:** esophageal squamous cell carcinoma, tumor microenvironment, cancer-associated fibroblasts, biglycan, TLR4

## Abstract

Esophageal squamous cell carcinoma (ESCC) is associated with poor prognosis due to aggressive invasion and therapy resistance. Cancer-associated fibroblasts (CAFs) are key stromal components that promote tumor progression; however, their specific roles in ESCC remain unclear. Using a direct co-culture model of ESCC cell lines (TE-9, -10, and -15) and mesenchymal stem cells (MSCs) to generate CAF-like cells, we identified biglycan (*BGN*) as a significantly upregulated gene in CAF-like cells via cDNA microarray analysis. Public single-cell RNA sequencing data also demonstrated elevated *BGN* expression in CAF clusters. We confirmed that CAF-like cells exhibited elevated BGN expression and secretion at both the mRNA and protein levels. Recombinant human BGN enhanced ESCC cell proliferation and migration by activating Erk and NF-κB signaling pathways, effects abrogated by TLR4 blockade. Furthermore, BGN promoted CAF marker expression in MSCs, M2-like macrophage polarization, and enhanced proliferation and migration abilities in both cell types. Immunohistochemical analysis of 66 ESCC tissues revealed that high stromal BGN expression correlated with greater tumor invasion, lymphatic invasion, and shorter disease-free survival. These findings indicate that CAF-derived BGN promotes ESCC progression via TLR4-mediated signaling and modulates stromal cell behavior, highlighting its potential as a prognostic biomarker and therapeutic target.

## 1. Introduction

Esophageal cancer (EC) is the seventh most frequently diagnosed cancer and the sixth leading cause of cancer-related mortality worldwide, with a dismal 5-year survival rate of approximately 20% [1,2]. EC is generally classified into esophageal adenocarcinoma and esophageal squamous cell carcinoma (ESCC), the latter of which accounts for nearly 85% of all [2]. ESCC is characterized by its aggressive local invasiveness, attributed to anatomical features such as the absence of a serosal layer, and by its propensity for lymphatic metastasis due to the dense lymphatic network within the submucosa of the esophagus [3]. Additionally, advanced ESCC frequently exhibits resistance to radiotherapy and chemotherapy, further contributing to its poor clinical outcomes [4,5].

In recent years, the tumor microenvironment has garnered increasing attention for its pivotal role in cancer progression [6]. The tumor microenvironment is composed not only of cancer cells but also of various stromal elements, including immune cells, endothelial cells, and fibroblasts. Among these cells, cancer-associated fibroblasts (CAFs) have emerged as central mediators of tumor progression, driving extracellular matrix (ECM) remodeling, secreting pro-tumorigenic factors, and regulating immune responses [7,8]. Rather than representing a homogeneous population, CAFs arise from diverse cellular origins, including resident fibroblasts and bone marrow-derived mesenchymal stem cells (MSCs), and exhibit marked heterogeneity [9,10]. Functionally distinct CAF subtypes, such as myofibroblastic CAFs (myCAFs), which contribute to ECM production and remodeling, and inflammatory CAFs (iCAFs), which secrete cytokines and modulate immune responses, have been described [11]. A wide range of CAF-derived molecules—including cytokines (e.g., interleukin-6 [IL6], transforming growth factor-β), growth factors, proteases, and matricellular proteins—have been shown to enhance tumor proliferation, invasion, angiogenesis, and immune evasion [12]. In ESCC, accumulating evidence indicates that CAF-derived soluble factors play crucial roles in promoting malignant phenotypes and therapy resistance [13,14,15,16,17]. Therefore, elucidating the mechanisms underlying ESCC progression and, in particular, understanding the interactions between cancer cells and CAFs, is a critical objective. Although the importance of CAF-derived factors has been widely reported, many aspects—such as the specific functions of different CAF subtypes and their links to ECM-associated proteins—remain unexplored.

Previously, we established a direct co-culture model involving ESCC cells and bone marrow-derived MSCs to investigate CAF–tumor interactions [16]. In this study, we focused on biglycan (BGN), which was significantly upregulated in CAF-like cells derived from this model. To address its potential role in ESCC progression, we performed transcriptomic profiling, in vitro functional assays, and histopathological analysis of clinical specimens. Specifically, we investigated whether CAF-derived BGN contributes to ESCC progression, the intracellular signaling cascades activated by BGN and its interaction with TLR4, as well as its potential effects on other stromal constituents, including fibroblasts and macrophages. Collectively, our findings reveal novel aspects of BGN biology within the ESCC tumor microenvironment and highlight its potential utility as both a prognostic biomarker and therapeutic target.

## 2. Results

### 2.1. Direct Co-Culture Induces the Upregulation of BGN Gene Expression and Protein Expression/Secretion Levels in CAF-like Cells, and BGN Promotes ESCC’s Malignant Phenotypes

A direct co-culture model of ESCC cell lines (TE-9, TE-10, and TE-15) with MSCs has been established previously to investigate the functional roles of CAFs within the ESCC tumor microenvironment [16]. We previously demonstrated that MSCs acquired CAF-like characteristics following direct co-culture, yielding ESCC cells termed TE co (TE-9, -10, and -15 co) and CAF-like cells designated as CAF9, CAF10, and CAF15, respectively. In contrast, monocultured cells were referred to as TE mono (TE-9, -10, and -15 mono) and MSC mono. Unlike our previous study [16], which focused on genes highly expressed in CAF9 relative to MSCs in direct versus indirect co-culture, the present study aimed to identify genes specifically associated with the functional transition of MSCs into CAFs. To this end, we selected genes that were upregulated in CAF9 relative to MSC mono, but showed low expression in both TE-9 mono and TE-9 co cells, and no significant increase in expression in TE-9 co compared to TE-9 mono. Specifically, candidate genes were filtered based on the following signal intensity and expression ratio criteria: (1) CAF9 > 1000; (2) TE-9 mono and TE-9 co < 100; (3) CAF9/MSC mono ratio > 2; and (4) TE-9 co/TE-9 mono ratio < 2. We identified eight genes by integrating these criteria using an Euler diagram (Figure 1A); these genes are listed in Table 1. Among them, *BGN* exhibited the highest expression in CAF9 and was therefore prioritized for further investigation.

The expression of BGN was evaluated using single-cell RNA-sequencing (scRNA-seq) data from the GSE160269 dataset. BGN expression was markedly higher in fibroblasts (Figure 1B). Unsupervised clustering of fibroblast scRNA-seq data identified 13 distinct clusters, which were subsequently annotated based on known markers. These clusters were annotated as five major subtypes: myCAF, iCAF, normal mucosa fibroblasts (NMF), normal activated fibroblasts (NAF), and vascular smooth muscle cells (VSMC). UMAP revealed distinct separation among these subtypes (Appendix A). Violin plot analysis showed significantly higher BGN expression in CAFs (myCAF and iCAF) compared to normal fibroblasts (NMF, NAF, and VSMC) (log_2_ fold change [log_2_FC] = 1.80, adjusted *p* < 0.001). Approximately 99% of CAFs expressed BGN, compared to 78% in normal fibroblasts (Figure 1C). When comparing myCAF and iCAF specifically, BGN expression was significantly higher in myCAF (log_2_FC = 0.29, adjusted *p* < 0.001), although both subtypes exhibited high overall expression frequencies (99.6% vs. 98.5%) (Appendix A).

Subsequently, BGN expression was assessed in CAF-like cells in our direct co-culture model. Quantitative reverse transcription polymerase chain reaction (qRT-PCR) and Western blot analyses revealed significant upregulation of BGN at both the mRNA and protein levels in CAF9, CAF10, and CAF15 compared to MSC mono (Figure 1D,E, Appendix A). Similarly, Western blot analysis of the culture supernatants demonstrated that the secretion levels of BGN protein were elevated in co-culture conditions for TE-9/CAF9, TE-10/CAF10, and TE-15/CAF15 compared to MSC mono (Figure 1F, Appendix A). To investigate the role of BGN in promoting ESCC malignancy, we assessed the effects of recombinant human BGN (rhBGN) on the proliferation and migration of ESCC cells. MTS assay revealed that rhBGN treatment significantly increased the proliferation of ESCC cells (Figure 1G). Consistently, the Transwell migration assay showed that rhBGN enhanced ESCC cell migration (Figure 1H). Moreover, treatment with a neutralizing antibody against BGN attenuated the CAF-induced increase in ESCC cell migration, indicating that CAF-derived BGN promotes ESCC cell migration (Figure 1I).

### 2.2. Knockdown of BGN in CAF-like Cells Attenuates CAF-Induced Proliferation and Migration of ESCC Cells

To further validate the functional contribution of CAF-derived BGN to ESCC progression, we performed knockdown experiments using two siRNAs targeting BGN (siBGN1 and siBGN2). The qRT-PCR and Western Blot analyses demonstrated that both siBGN1 and siBGN2 efficiently reduced BGN expression at the mRNA and protein levels in CAF9, CAF10, and CAF15 compared to negative control-treated CAF-like cells (Figure 2A,B, Appendix A). Consistently, Western Blot analysis of the co-culture supernatants showed that BGN secretion was markedly suppressed in TE/CAF direct co-culture systems following BGN knockdown (Figure 2C, Appendix A). To assess the functional relevance of endogenous BGN, we conducted MTS assays using conditioned media obtained from TE/CAF direct co-cultures treated with siRNAs. The CAF-induced enhancement of ESCC cell proliferation was abolished by BGN knockdown (Figure 2D). Similarly, Transwell migration assays revealed that the CAF-mediated increase in ESCC cell migration was also attenuated by BGN knockdown (Figure 2E). These findings indicate that endogenous CAF-derived BGN plays a crucial role in promoting the proliferation and migration of ESCC cells.

### 2.3. BGN Promotes the Proliferation and Migration of ESCC Cells Through the Erk and NF-κB Signaling Pathways

BGN has been shown to enhance tumor malignancy via activation of the NF-κB signaling pathway [18,19,20]. In this study, phosphorylation of NF-κB was increased in ESCC cells following direct co-culture compared to monocultured cells (Appendix A). To further investigate the signaling pathways activated by BGN, rhBGN was added to ESCC cells, and phosphorylation of signaling proteins were assessed at 10, 30, and 60 min post-treatment. rhBGN treatment led to increased phosphorylation of both NF-κB and Erk, indicating activation of these pathways (Figure 3A, Appendix A). Furthermore, treatment of ESCC cells with Erk (PD98059) or NF-κB (Bay11-7082) pathway inhibitors attenuated the BGN-induced enhancement of proliferation and migration (Figure 3B,C). These results confirm the involvement of Erk and NF-κB signaling in mediating BGN’s effects on ESCC cells.

### 2.4. BGN Promotes Proliferation and Migration Through Its Receptor TLR4

BGN has been reported to bind both TLR2 and TLR4 in various cell types, including cancer cells [19,20,21,22,23]. Previous studies have reported that TLR4 is overexpressed in ESCC, that its overexpression is associated with poor prognosis, and that it promotes cancer cell proliferation through the TLR4 signaling pathway [24,25,26]. In contrast, few studies have demonstrated high expression of TLR2 in ESCC. Furthermore, analysis of public data from 81 ESCC cases using the Kaplan–Meier Plotter revealed that recurrence-free survival was significantly shorter in the TLR4 high-expression group than in the low-expression group (*p* = 0.026) (Appendix A). In contrast, no significant differences were observed in overall survival and recurrence-free survival between the TLR2 high- and low-expression groups (*p* = 0.14 and *p* = 0.27, respectively) (Appendix A). Collectively, these findings led us to focus on TLR4 in the subsequent analyses of this study. Western Blot analysis confirmed TLR4 expression in all ESCC cell lines (Figure 4A, Appendix A). Immunofluorescence staining revealed colocalization of BGN and TLR4 in all ESCC cell lines treated with rhBGN (Figure 4B). Furthermore, the rhBGN-induced proliferation and migration of ESCC cells were abolished by treatment with a neutralizing antibody against TLR4, as shown by MTS and Transwell migration assays (Figure 4C,D). These findings indicate that BGN functions as a ligand for TLR4 on ESCC cells.

### 2.5. BGN Contributes to the Activation Phenotypes of MSCs and Macrophages

To investigate the role of BGN-expressing CAFs in the tumor microenvironment, CAFs in the scRNAseq dataset were classified into CAF_BGN_High and CAF_BGN_Low groups based on BGN expression levels in a cluster of CAFs (scaled expression > 3 or ≤3, respectively) in the public database of ESCC tissues. CellChat analysis revealed that CAF_BGN_High cells exhibited strong outgoing signals toward both myeloid and epithelial cells, as well as prominent autocrine signaling within the CAF_BGN_High population itself (Figure 5A). These findings suggest that BGN-expressing CAFs may influence macrophages—a major myeloid subset—as well as reinforce their own activation state through autocrine signaling.

To further explore these possibilities, we conducted stimulation experiments by adding rhBGN to MSCs or macrophages. The MTS assay showed that rhBGN-treated MSCs had increased proliferation (Figure 5B). The Transwell migration assay demonstrated increased migration in rhBGN-treated MSCs (Figure 5C). Western Blot analysis demonstrated that the protein expression of fibroblast activation protein (FAP) and α-smooth muscle actin (αSMA), recognized as CAF markers, was upregulated in MSCs treated with rhBGN for 24 h, whereas IL6 protein expression remained unchanged. TLR4 expression was detected in MSCs; however, the addition of rhBGN did not increase the phosphorylation of NF-κB and Erk (Figure 5D, Appendix A). Furthermore, rhBGN-induced proliferation and migration of MSCs remained unchanged even in the presence of a TLR4-neutralizing antibody, as shown by MTS and Transwell migration assays (Figure 5E,F). Moreover, similar experiments were performed in macrophages. The rhBGN-treated macrophages also showed increased proliferation and migration (Figure 5G,H). In addition, Western Blot analysis revealed that rhBGN-treated macrophages upregulated the expression of CD163 and CD206, known M2 macrophage markers, and increased NF-κB phosphorylation. TLR4 expression was also detected; however, Erk phosphorylation was not induced (Figure 5I, Appendix A). The rhBGN-induced proliferation and migration of macrophages were attenuated by treatment with a neutralizing antibody against TLR4, as shown by MTS and Transwell migration assays (Figure 5J,K). Furthermore, treatment with the NF-κB pathway inhibitor Bay 11-7082 also attenuated the rhBGN-induced enhancement of proliferation and migration (Figure 5L,M).

### 2.6. BGN Expression Is Elevated in CAFs Within ESCC Tissues and Is Correlated with Tumor Progression in Disease-Free Survival

To investigate the clinical relevance of BGN expression, immunohistochemical staining was performed on 66 human ESCC tissue samples. BGN expression in ESCC tissues was predominantly observed in the stroma, not within tumor nests. Therefore, staining intensity was evaluated in the stromal compartment at the tumor invasive front (Figure 6A). Based on staining intensity, patients were stratified into low-expression (n = 31) and high-expression (n = 35) groups (Figure 6A). Kaplan–Meier survival analysis revealed that disease-free survival was significantly shorter in the BGN high-expression group compared to the BGN low-expression group (*p* < 0.01) (Figure 6B). In contrast, no significant differences in overall survival or cause-specific survival between the two groups (*p* = 0.36 and *p* = 0.10, respectively) (Figure 6B).

In addition, high BGN expression was significantly associated with tumor invasion depth (*p* < 0.001), lymphatic vessel invasion (*p* = 0.001), and increased expression of CAF markers, αSMA (*p* = 0.003) and FAP (*p* = 0.022), and macrophage markers, CD68 (*p* = 0.049) and CD163 (*p* = 0.007), in ESCC tissues (Table 2).

## 3. Discussion

Various factors secreted by CAFs have been shown to contribute critically to tumor progression [30,31,32]. Our previous studies showed that both indirect [13,14,15] and direct co-culture [16,17] with CAFs promotes malignant phenotypes of ESCC cells. Analysis of the indirect co-culture model revealed that multiple soluble factors (CCL2, IL-6, PAI-1, and IGFBP2) promote the progression of ESCC [13,14,15]. More recently, periostin was identified as a CAF-derived soluble factor involved in ESCC progression using a direct co-culture model, highlighting differences from the indirect co-culture model [16]. In this study, we identified BGN as a new CAF-derived soluble factor by focusing on genes that showed low expression in cancer cells but showed specific expression increases in CAFs in a direct co-culture model. BGN has been reported to act as a secreted protein with pro-tumorigenic effects in several cancers [19,33,34], and in this study, we investigated its functional role in ESCC progression.

BGN, a member of the small leucine-rich proteoglycan (SLRP) family, is a structural component of the ECM predominantly found in connective tissues such as bone and tendon, where it contributes to collagen fibrillogenesis and tissue integrity [35,36,37]. While traditionally regarded as a structural molecule, recent studies have highlighted its active physiological functions as a secreted “matrikine” involved in regulating inflammation and immune responses [23]. Following tissue injury or cellular stress, BGN is released from the ECM and acts as a danger-associated molecular pattern, thereby activating innate responses. Specifically, BGN induces CXCL13 expression in macrophages and dendritic cells [22]. BGN has also been shown to activate the NLRP3 inflammasome, thereby promoting the maturation and secretion of IL-1β, indicating its role in both acute and chronic inflammation [21]. Its pro-inflammatory activity has been implicated in the pathogenesis of autoimmune diseases, fibrotic disorders, and cardiovascular conditions. In recent years, its role within the tumor microenvironment has gained increasing attention. In several solid malignancies, including pancreatic cancer [38], breast cancer [39], colorectal cancer [40], and head and neck squamous cell carcinoma [41], BGN has been reported to promote tumor cell phenotypes, such as proliferation, migration, and immune suppression, thereby contributing to tumor progression. While BGN has been extensively studied in other cancers, its expression and functional involvement in ESCC have remained largely unexplored. To address this knowledge gap, we investigated the potential role of BGN in ESCC development and progression.

Various receptors have been identified as binding partners of BGN; among them, TLRs have been most frequently reported in the context of inflammation and the tumor microenvironment. In macrophages, BGN acts as an endogenous ligand for TLR2 and TLR4, rapidly activating innate immune signaling pathways such as p38, Erk, and NF-κB. This activation leads to the expression of pro-inflammatory mediators, including TNF-α and macrophage inflammatory protein-2 (MIP-2) [23]. In gastric endothelial cells, BGN enhances NF-κB binding to the HIF-1α promoter via TLR2 and TLR4 signaling, resulting in increased VEGF expression and enhanced migration of gastric cancer cells [20]. Other receptors, such as the type I insulin-like growth factor receptor (IGF-IR) [42] and low-density lipoprotein receptor-related protein 6 (LRP6) [43], have also been implicated in BGN-mediated signaling; however, TLR2 and TLR4 remain the most widely studied and recognized BGN receptors. In this study, we identified TLR4 as the key receptor mediating BGN-induced activation of the Erk and NF-κB pathways in ESCC, providing novel mechanistic insight into BGN’s role in this malignancy.

CAFs promote tumor progression in part by secreting soluble factors that act on neighboring cells via paracrine signaling [30,31,32]. In ESCC, several studies have demonstrated that CAF-derived factors contribute to tumor progression [13,14,15,16]. In the present study, we focused on BGN, which was significantly upregulated in CAFs compared to MSCs in our previously established direct co-culture model, and explored its functional role in ESCC. Previous reports have implicated CAF-derived BGN in tumor progression across several cancer types. For instance, in breast cancer, high BGN expression is negatively correlated with CD8^+^ T cell infiltration and is associated with an immunosuppressive tumor microenvironment [34]. In colorectal cancer, BGN downregulation in CAFs markedly reduced cancer cell migration and proliferation [33]. Consistent with these findings, we found that CAF-derived BGN promotes proliferation and migration of ESCC cells. Moreover, our findings suggest that BGN exerts not only paracrine effects within tumor–stromal interactions but also autocrine effects on CAFs, and may influence macrophages in the tumor microenvironment. In this study, we observed that BGN functioned as a ligand for TLR4 and activated the NF-κB signaling pathway, thereby enhancing proliferation and migration in macrophages. This observation is consistent with previous studies showing that BGN binds to TLR4 and functions as its ligand to activate the NF-κB signaling pathway in macrophages, thereby inducing the expression of inflammatory mediators [23]. In thyroid cancer, this BGN–TLR4 interaction also promotes macrophage polarization toward the M2 phenotype [44]. This supports the concept that BGN contributes to the establishment of an immunosuppressive tumor microenvironment through the modulation of macrophage function. In contrast, BGN acted on MSCs to induce proliferation and migration, but the intracellular signaling mechanisms in these phenotypes remain unclear. Additionally, we observed despite expressing TLR4, rhBGN stimulation did not alter NF-κB or Erk phosphorylation in MSCs. Furthermore, the rhBGN-induced effects in MSCs were not inhibited by treatment with a TLR4 neutralizing antibody. These results suggest that BGN-induced activation of MSCs may occur through a TLR4-independent pathway. The BGN promotes osteogenic differentiation and proliferation in MSCs [45]; however, the receptor systems and downstream signaling pathways mediating its effects in the microenvironment remain unelucidated. Given that BGN can interact with multiple receptors, including IGF-IR [42] and LRP6 [43], the alternative receptor-mediated pathways may be involved in the observed effects. Further studies are warranted to elucidate the molecular mechanisms by which BGN activates MSCs and induces their differentiation into CAFs. Taken together, although CAF-derived autocrine signaling [46,47] and BGN’s ability to modulate macrophages [48] have been reported in other cancer types, these mechanisms have not yet been described in ESCC. Our study thus expands the current understanding of tumor–stromal interactions in ESCC and highlights BGN as a key mediator within its tumor microenvironment.

CAFs are known to comprise distinct functional subtypes. According to Öhlund et al. [11], iCAFs secrete cytokines such as IL6 and support immune modulation and tumor progression. In contrast, myCAFs, marked by high αSMA expression, contribute to ECM production and structural support near tumor cells. In this study, we confirmed that BGN is highly expressed in both iCAFs and myCAFs, which is consistent with previous findings [49]. BGN has been reported to function both as a structural ECM component [35,50] and as an inducer of inflammatory cytokines [21,22], suggesting that it may exert dual function. Given that BGN upregulation was observed across major CAF subsets, our findings suggest that BGN represents a common CAF-associated molecule rather than a subtype-restricted marker. This broad expression pattern raises the possibility that BGN could serve as a therapeutic target for CAFs across the ESCC tumor microenvironment.

Several limitations of this study should also be acknowledged: First, this study did not include in vivo experiments. Further investigations using animal models are needed to evaluate the physiological relevance of BGN in tumor progression and immune modulation. Second, although rhBGN stimulation enhanced proliferation, migration, and CAF marker expression in MSCs, it did not activate NF-κB or Erk signaling, and these effects were not affected by TLR4 neutralization. These findings suggest that BGN may act through alternative receptors or signaling mechanisms in MSCs, which remain to be elucidated in future studies. Finally, while this study focused on TLR4 as the primary receptor for BGN, the potential involvement of other known or unknown receptors, including TLR2, remains unexplored and warrants further analysis.

## 4. Materials and Methods

### 4.1. Cell Lines and Cell Culture

Three human ESCC cell lines—TE-9, TE-10, and TE-15—were obtained from the RIKEN BioResource Center (Tsukuba, Japan). Cells were maintained in RPMI-1640 medium (FUJIFILM Wako Pure Chemical Corporation, Osaka, Japan) supplemented with 10% fetal bovine serum (FBS; Sigma-Aldrich, St. Louis, MO, USA) and 1% Penicillin-Streptomycin-Amphotericin B (antibiotics; FUJIFILM Wako Pure Chemical Corporation) at 37 °C in a humidified 5% CO_2_ incubator. Human bone marrow-derived MSCs (PCS-500-012, ATCC, Manassas, VA, USA) were cultured in low-glucose Dulbecco’s Modified Eagle Medium (DMEM; FUJIFILM Wako Pure Chemical Corporation) containing 10% FBS and 1% antibiotics. CD14-positive monocytes were isolated from healthy volunteer peripheral blood using anti-CD14 magnetic beads (#130-050-201, Miltenyi Biotec, Bergisch Gladbach, Germany) on the autoMACS Pro Separator (Miltenyi Biotec). The enriched monocytes (5 × 10^5^ cells/well) were cultured in RPMI-1640 medium with 10% FBS supplemented with 10 ng/mL recombinant human macrophage colony-stimulating factor (R&D Systems, Minneapolis, MN, USA) and 1 ng/mL recombinant human granulocyte-macrophage colony-stimulating factor (R&D Systems), and incubated for 6 days to generate macrophages [51,52].

### 4.2. Direct Co-Culture Model

The direct co-culture model was established in a previous study [16]. Briefly, MSCs (3 × 10^5^ cells) were seeded into 100 mm culture dishes, and ESCC cells (2 × 10^5^ cells) were added to the same dishes 3 h later. Monocultures for each cell were prepared in parallel under identical conditions. After 4 days of culture, cells were washed with phosphate-buffered saline (PBS; FUJIFILM Wako Pure Chemical Corporation) and harvested using trypsin-EDTA (FUJIFILM Wako Pure Chemical Corporation). ESCC cells were separated by positive selection using anti-CD326 (EpCAM) microbeads (130-061-101, Miltenyi Biotec) with the autoMACS Pro Separator, while MSCs were obtained as EpCAM-negative fractions by negative selection.

### 4.3. cDNA Microarray Analysis

Gene expression profiling was previously conducted using the 3D-Gene Human Oligo Chip 25k (Toray Industries, Tokyo, Japan). Comparisons were made between monocultured TE-9 cells and TE-9 cells co-cultured with MSCs, and between monocultured MSCs and MSCs co-cultured with TE-9 cells (referred to as CAF9, as described later) [16,17]. The datasets have been deposited in the Gene Expression Omnibus (GEO) under accession numbers GSE274064 and GSE244020, respectively.

### 4.4. Single-Cell RNA Sequencing (scRNA-Seq)

Publicly available scRNA-seq data from ESCC tissues were obtained from the GEO under accession number GSE160269 [53]. This dataset comprises an analysis of 208,659 single-cell transcriptomes derived from 60 human ESCC cases. Raw UMI count data were processed in R (version 4.5.0) and converted into a Seurat object using the Seurat package (version 4.3.0). Cells were filtered based on standard quality control metrics. The dataset was normalized using the NormalizeData function, and highly variable genes were identified with the FindVariableFeatures function. The data were then scaled using the ScaleData function, and principal component analysis (PCA) was performed using the RunPCA function. The first 20 principal components were used for dimensionality reduction and downstream analysis. Cell clustering was performed using the FindNeighbors and FindClusters functions with a resolution parameter of 0.5. Uniform Manifold Approximation and Projection (UMAP) was used for two-dimensional visualization using the RunUMAP function. Clusters were visualized using the DimPlot function. Clusters were manually annotated based on the expression of established marker genes for fibroblast subtypes, including myCAF, iCAF, NMF, NAF, and VSMC. Marker gene expression across fibroblast clusters was visualized using the DotPlot and FeaturePlot functions to assess expression patterns and distribution. Fibroblasts were grouped into two major categories: CAFs (myCAF and iCAF) and normal fibroblasts (NMF, NAF, and VSMC). BGN expression was compared between CAFs and normal fibroblasts using violin plots generated with the VlnPlot function. Statistical differences in gene expression were assessed using the Wilcoxon rank-sum test implemented in the FindMarkers function. To evaluate intercellular communication between CAFs and other cell types, cell–cell interaction analysis was performed using the CellChat package (version 2.1.2). Before CellChat analysis, data integration was conducted using the FindIntegrationAnchors and IntegrateData functions in Seurat to correct for batch effects between the GSE160269_CD45neg_UMIs.txt and GSE160269_CD45pos_UMIs.txt datasets. This integration enabled the analysis of fibroblast-centered intercellular communication within the tumor microenvironment [54]. From the integrated Seurat object, major cell populations—including fibroblast subsets, immune cells, endothelial cells, and epithelial cells—were identified, and a CellChat object was constructed based on the defined clusters. The inferred cell–cell communication network was visualized using a circular plot generated with the netVisual_circle function.

### 4.5. Quantitative Real-Time PCR (qRT-PCR)

Total RNA was extracted using the RNeasy Mini Kit (Qiagen, Hilden, Germany) and quantified on a NanoDrop Lite spectrophotometer (Thermo Fisher Scientific, Waltham, MA, USA). cDNA was synthesized, and gene expression levels of *BGN* and *GAPDH* were assessed via SYBR Green-based qRT-PCR on a StepOne Real-Time PCR System (Applied Biosystems, Foster City, CA, USA). Ct values were normalized to *GAPDH* using the comparative Ct method [55]. Primer sequences were as follows: *BGN*, 5′-GGTCTGAAGTCTGTGCCCAA-3′ (forward) and 5′-GAGCTCGGAGATGTCGTTGT-3′ (reverse); *GAPDH*, 5′-GCACCGTCAAGCCTGAGAAT-3′ (forward) and 5′-ATGGTGGTCAAGACGCCAGT-3′ (reverse).

### 4.6. Western Blotting

Cells were lysed on ice in lysis buffer containing 50 mM Tris-HCl (pH 7.5), 125 mM NaCl, 5 mM EDTA, 0.1% Triton X-100, and protease/phosphatase inhibitors (Sigma-Aldrich), as described previously [13,14,15,16,17]. Conditioned media were collected from 6 days monocultures or co-cultures to detect secreted proteins in supernatants. Protein concentration for the extracted protein was measured via NanoDrop Lite. Lysates were resolved on 5–20% SDS-PAGE (FUJIFILM Wako Pure Chemical Corporation) and transferred to PVDF membranes using the iBlot2 system (Invitrogen, Carlsbad, CA, USA). After blocking with 5% skim milk in TBS, membranes were incubated overnight at 4 °C with primary antibodies, followed by HRP-conjugated secondary antibodies. Protein bands were visualized with ImmunoStar reagents (FUJIFILM Wako Pure Chemical Corporation) and imaged using an ImageQuant LAS4000 mini (FUJIFILM, Tokyo, Japan). The quantification of bands was performed using ImageJ software version 1.53t (National Institutes of Health, Bethesda, MD, USA, last accessed on 18 August 2025). The primary antibodies used as follows: rabbit BGN (#16409-1-AP, R&D Systems), rabbit phosphorylated Erk1/2 (pErk; #9101, Cell signaling Technology; CST, Danvers, MA, USA), rabbit total Erk1/2 (tErk; #9102, CST), rabbit phosphorylated NF-κB p65 (pNF-κB; #3033, CST), rabbit total NF-κB p65 (tNF-κB; #8242, CST), mouse TLR4 (#sc-293072, Santa Cruz Biotechmology, Dallas, TX, USA), sheep FAP (#AF3715, R&D Systems), rabbit αSMA (#ab5694, Abcam, Cambridge, UK), rabbit IL-6 (#ab6672, Abcam), mouse CD163 (#NCL-L-CD163; Leica Biosystems, Wetzlar, Germany), mouse CD206 (#sc-376108, Santa Cruz), and rabbit β-actin (#4970, CST) antibodies. Secondary antibodies included horseradish peroxidase (HRP)-conjugated donkey anti-rabbit (#NA934V, Cytiva, Marlborough, MA, USA), anti-mouse (#NA931V, Cytiva), and anti-sheep (#ab6900, Abcam) IgGs.

### 4.7. Cell Proliferation Assay

ESCC cells (1 × 10^4^/well) were seeded in serum-free RPMI-1640 medium in 96-well plates. Treatments applied to ESCC cells included rhBGN (#2667-CM; 100 ng/mL, R&D Systems) with or without PD98059 (Erk signaling inhibitor; 5 μM, CST), Bay 11-7082 (NF-κB signaling inhibitor: 1 μM, Sigma), DMSO vehicle (5 μM), TLR4 neutralizing mouse antibody (#NB100-56723; 1 μg/mL, Novus Biologicals, Centennial, CO, USA), or control mouse IgG (sc-2025; 1 μg/mL, Santa Cruz). Cell proliferation was assessed 48 h after treatment using the CellTiter 96^®^ AQueous One Solution Reagent (Promega, Madison, WI, USA), based on the MTS assay [3-(4,5-dimethylthiazol-2-yl)-5-(3-carboxymethoxyphenyl)-2-(4-sulfophenyl)-2H-tetrazolium]. The absorbance at 492 nm was measured using a microplate reader (Infinite 200 PRO, Tecan, Männedorf, Switzerland). Similar protocols were used for MSCs and macrophages with the addition of TLR4 neutralizing goat antibody (AF1478; 1 μg/mL, R&D Systems) or control goat IgG (AB-108-C; 1 μg/mL, R&D Systems) were used.

### 4.8. Transwell Migration Assay

Cells were seeded into Transwell inserts (8.0 µm pore size; BD Falcon, BD Biosciences, San Jose, CA, USA), each with 300 µL of serum-free medium. The lower chamber contained 800 µL of the corresponding medium. Treatments conditions mirrored those in the proliferation assays: rhBGN with or without PD98059 (5 μM), Bay 11-7082 (1 μM), DMSO (5 μM), TLR4 mouse neutralizing antibody (1 μg/mL), or control mouse IgG (1 μg/mL). In one variation, ESCC cell lines (TE-9, TE-10, TE-15) and MSCs (CAF9, CAF10, CAF15) cultured under direct co-culture conditions for 96 h were harvested and used together in the migration assay. After 96 h of culture, ESCC cells and MSCs were harvested and seeded together at a total of 5 × 10^4^ cells/well into the lower chamber containing 0.1% FBS medium, with or without a BGN neutralizing rabbit antibody (220 ng/mL, R&D Systems) or an equivalent concentration of control rabbit IgG (#PM035; 220 ng/mL, MBL, Japan). After 48 h, migrated cells under the Transwell membrane were stained with Diff-Quik (Sysmex, Kobe, Japan) and counted. Similar assays were performed for MSCs and macrophages under serum-free conditions with the addition of TLR4 neutralizing goat antibody (1 μg/mL) or control goat IgG (1 μg/mL).

### 4.9. Knockdown of BGN in the Direct Co-Culture Model

For BGN knockdown in CAF-like cells, the direct co-culture between MSCs and ESCC cell lines, as previously described, was performed for 4 days with siRNA. In brief, on day 1, after MSCs were seeded, ESCC cell lines were added to the same dish once they had adhered and then transfected with two independent siRNAs targeting human BGN (siBGN1 and siBGN2; 20 pmol each), custom-synthesized based on the human BGN target sequence (Nippon Gene Co., Ltd., Toyama, Japan), or with MISSION siRNA Universal Negative Control #1 (siNC; 20 pmol; Sigma-Aldrich) as a negative control. The siRNA sequences were as follows: siBGN1 sense, 5′-UCUGAAGUCUGUGCCCAAATT-3′, and antisense, 5′-UUUGGGCACAGACUUCAGATT-3′; siBGN2 sense, 5′-GCCAUUCAUGAUGAACGAUTT-3′, and antisense, 5ʹ-AUCGUUCAUCAUGAAUGGCTT-3ʹ. Transfection was performed using Lipofectamine RNAiMAX (Invitrogen) according to the manufacturer’s instructions. After 4 days of direct co-culture with siRNA transfection, CAF-like cells were isolated from the co-culture as described above and used for subsequent experiments.

### 4.10. Immunofluorescence

TE-9, TE-10, and TE-15 cells were cultured in serum-free RPMI-1640 for 8 h with or without rhBGN treatment. Cells (3 × 10^4^) were then seeded into Lab-Tek II 4-well chamber slides (Thermo Fisher) and incubated overnight. Cells were fixed with 4% paraformaldehyde at room temperature for 10 min. After washing, cells were incubated overnight at 4 °C with rabbit anti-BGN (1:500; R&D Systems) and mouse anti-TLR4 (1:50; Santa Cruz Biotechnology) antibodies. Alexa Fluor 488-conjugated donkey anti-rabbit (1:200; #711-545-152; Jackson ImmunoResearch Laboratories, West Grove, PA, USA) and Cy3-conjugated donkey anti-mouse secondary antibodies (1:200; #715-165-150; Jackson ImmunoResearch) were applied (protected from light), followed by nuclei counterstaining with DAPI (1:1000; #340-07971; Wako). Imaging was performed with a Zeiss LSM 700 confocal microscope (Carl Zeiss, Oberkochen, Germany).

### 4.11. Tissue Samples

Archived surgical ESCC specimens from 66 patients treated at Kobe University Hospital between 2005 and 2010 were analyzed. Patients with prior chemo/radiotherapy were excluded. Tissue samples were formalin-fixed and paraffin-embedded. Pathological staging followed the 10th edition of the Japanese Classification of Esophageal Cancer and the 7th UICC TNM classification. This study was approved by the Kobe University Institutional Review Board (B210103) and conducted with informed consent in accordance with the Declaration of Helsinki.

### 4.12. Immunohistochemistry

Four-micrometer tissue sections were stained using the BOND Polymer Refine Detection Kit with the BOND-MAX automated system (Leica Biosystems, Nubloch, Germany). Staining was performed using an anti-BGN polyclonal antibody (#16409-1-AP; Proteintech) at a dilution of 1:2000. Staining at the tumor invasive front was scored as follows: negative, 0; weak, 1; moderate, 2; strong, 3. Cases with scores of 0 or 1 were classified as low BGN expression, while scores of 2 or 3 were considered high expression. Two blinded observers (a pathologist (Y.-i.K.) and a surgeon (H.Y)) evaluated all samples.

### 4.13. Bioinformatics Analysis

Survival analysis was performed using the Kaplan–Meier Plotter [56] (https://kmplot.com/analysis/index.php?p=home; accessed 18 August 2025) based on mRNA expression levels of *TLR2* and *TLR4* in 81 ESCC patients stratified into high- and low-expression groups. Patients were divided into high-and low-expression groups based on the median expression value, and overall survival and recurrence-free survival were evaluated.

### 4.14. Statistical Analysis

Each experiment was performed in triplicate and independently repeated three times. Data are presented as mean ± standard error of the mean. Comparisons among groups were analyzed using Student’s *t*-test or the Tukey–Kramer test. Associations with clinicopathological factors were assessed using the chi-squared test. Kaplan–Meier survival curves were compared using the log-rank test. All analyses were conducted using GraphPad Prism version 6.0 (GraphPad Software, Boston, MA, USA), with statistical significance set at *p* < 0.05.

## 5. Conclusions

In conclusion, our study demonstrates that BGN is significantly upregulated in CAFs within the ESCC tumor microenvironment and promotes the proliferation and migration of ESCC cells through activation of the Erk and NF-κB pathways via TLR4 signaling. In addition to its paracrine effects on tumor cells, BGN exerted autocrine effects on fibroblasts, promoting proliferation, migration, and CAF transition. Moreover, in macrophages, BGN promoted proliferation and migration, and induced M2-like polarization by activating the NF-κB signaling pathway through TLR4, suggesting its multifaceted role in modulating the tumor microenvironment (Figure 7). These findings support the potential of BGN as both a therapeutic target and a prognostic biomarker in ESCC. Given that BGN functions through the TLR4–Erk/NF-κB axis, therapeutic strategies such as TLR4 blockade, pathway inhibition, or direct suppression of BGN expression may help mitigate its protumorigenic effects. Further exploration of BGN-targeting approaches—such as small molecules, neutralizing antibodies, or RNA-based therapeutics—may contribute to the development of novel treatments for ESCC.

## Figures and Tables

**Figure 1 ijms-26-12024-f001:**
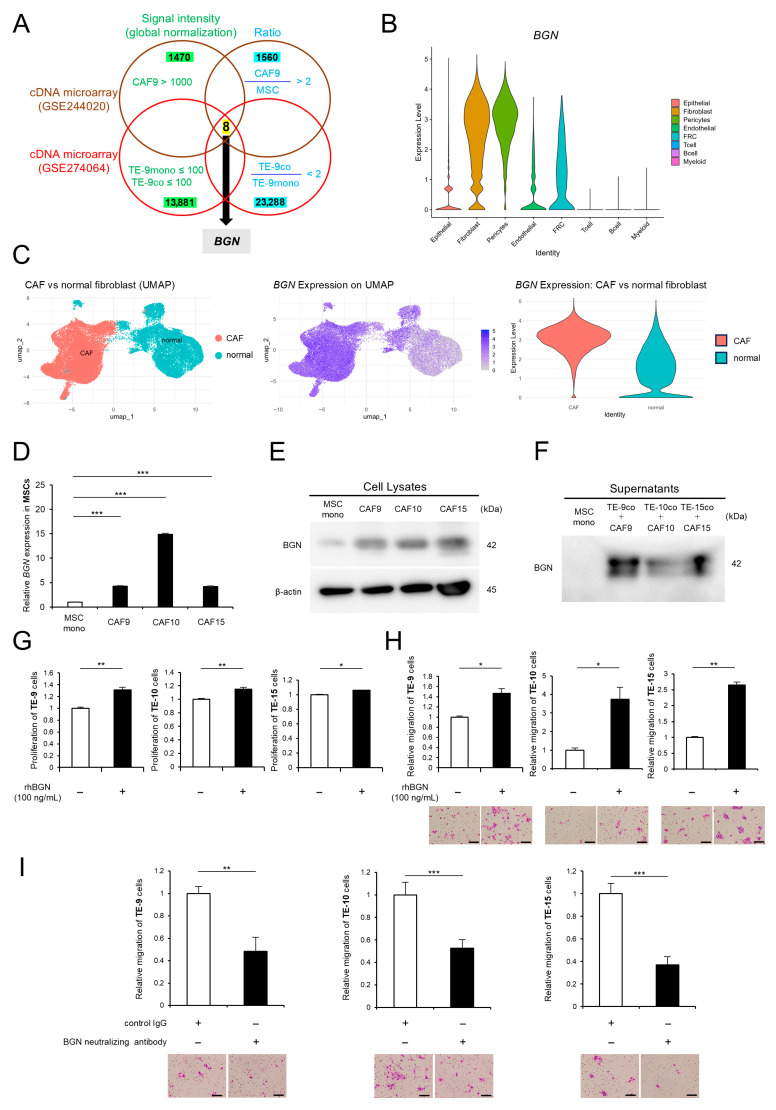
Biglycan (BGN), highly expressed in cancer-associated fibroblasts (CAFs), promotes the proliferation and migration of esophageal squamous cell carcinoma (ESCC) cells. (**A**) Euler diagram showing the overlap between genes meeting the signal intensity value criteria (CAF9 > 1000, TE-9 mono ≤ 100, and TE-9 co ≤ 100) and the expression ratio criteria (CAF9/MSC mono > 2 and TE-9 co/TE-9 mono < 2) in the cDNA microarray analysis (GSE244020 and GSE274064). Eight overlapping genes were identified, with BGN showing the highest signal intensity value in CAF9. (**B**) Violin plot showing *BGN* expression across cell types based on single-cell RNA sequencing (scRNAseq) data from the GSE160269 dataset. (**C**) Uniform Manifold Approximation and Projection (UMAP) and *BGN* expression in fibroblasts based on scRNAseq data. Violin plot visualization shows significantly higher *BGN* expression in CAFs compared to normal fibroblasts. (**D**–**F**) The mRNA expression, protein expression, and secreted protein levels of BGN in mesenchymal stem cell (MSC) mono and CAFs were compared using quantitative reverse transcription polymerase chain reaction (qRT-PCR) (**D**) and Western blotting of cell lysates and supernatants (**E**,**F**), respectively. (**G**) The effects of recombinant human BGN (rhBGN) (100 ng/mL) on the proliferation of ESCC cells were evaluated using the MTS assay. (**H**,**I**) The effect of rhBGN (100 ng/mL) on ESCC cell migration (**H**) and the effect of BGN neutralizing antibody (220 ng/mL) on CAF-mediated ESCC cell migration (**I**) were assessed using the Transwell migration assay. In (**H**), ESCC cells were seeded in the upper chamber; in (**I**), ESCC cells were seeded in the upper chamber, while direct co-culture of ESCC cells and MSCs was performed in the lower chamber. The rhBGN (**H**) and BGN neutralizing antibody (220 ng/mL) or control immunoglobulin G (IgG; 220 ng/mL) (**I**) were added to the lower chamber. Migrated cells were counted in five representative microscopic fields after 48 h, and representative images are shown below the graphs. The data are presented as the mean ± standard error of the mean (SEM) from three independent experiments (**D**,**G**–**I**). * *p* < 0.05, ** *p* < 0.01, *** *p* < 0.001. Scale bars: 100 μm (**H**,**I**). FRC, fibroblastic reticular cells.

**Figure 2 ijms-26-12024-f002:**
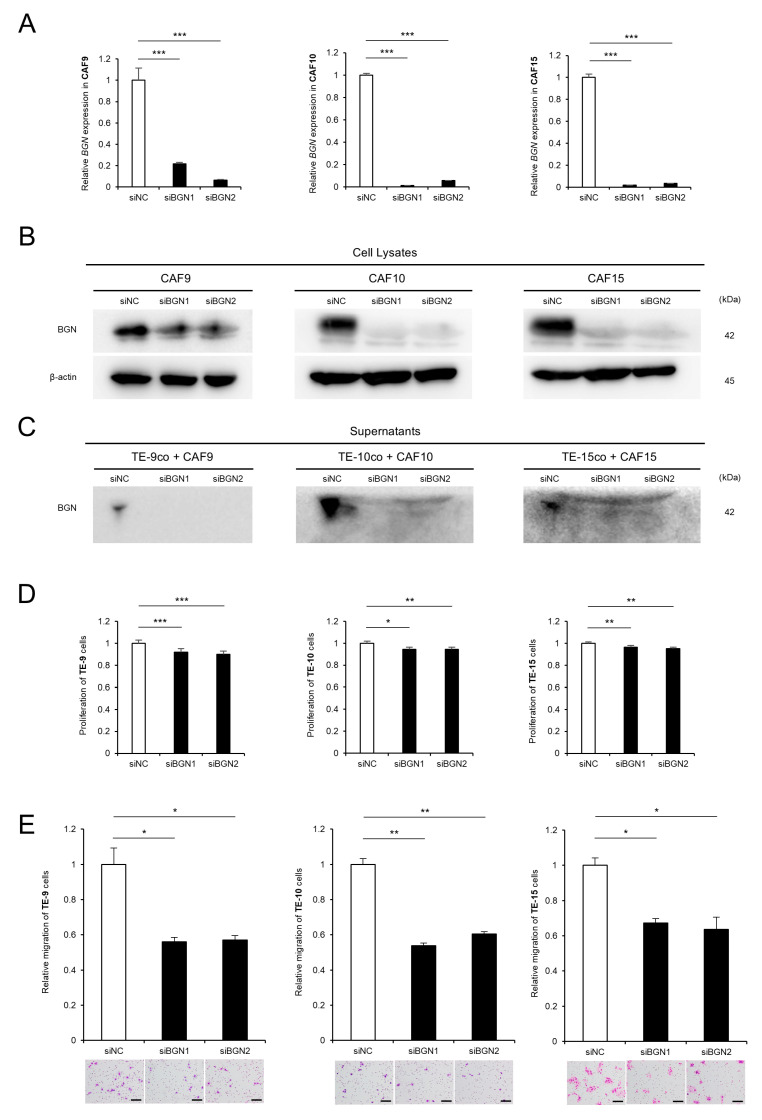
Knockdown of biglycan (BGN) in CAF-like cells suppresses esophageal squamous cell carcinoma (ESCC) cell proliferation and migration. (**A**,**B**) CAF9, CAF10, and CAF15 cells were transfected with two siRNA against BGN (siBGN1 and siBGN2), or negative control siRNA (siNC). The knockdown efficiency of BGN at the mRNA and protein levels was confirmed by quantitative reverse transcription polymerase chain reaction (qRT-PCR) (**A**) and Western Blot analysis (**B**). (**C**) Western Blot analysis of culture supernatants demonstrated that BGN secretion was markedly reduced in TE/CAF direct co-culture systems following siBGN1 and siBGN2 transfection compared to siNC. (**D**,**E**) Conditioned media collected from TE/CAF direct co-culture systems with siRNA were used to evaluate the effects of BGN knockdown on ESCC cell proliferation and migration. MTS assay (**D**) and Transwell migration assay (**E**) showed that the CAF-induced enhancement of ESCC cell proliferation and migration was abolished by siBGN1 and siBGN2 compared to siNC. Migrated cells were counted in five representative microscopic fields after 48 h, and representative images are shown below the graphs. Data represent the mean ± standard error of the mean (SEM) from three independent experiments. * *p* < 0.05, ** *p* < 0.01, *** *p* < 0.001. Scale bars: 100 μm.

**Figure 3 ijms-26-12024-f003:**
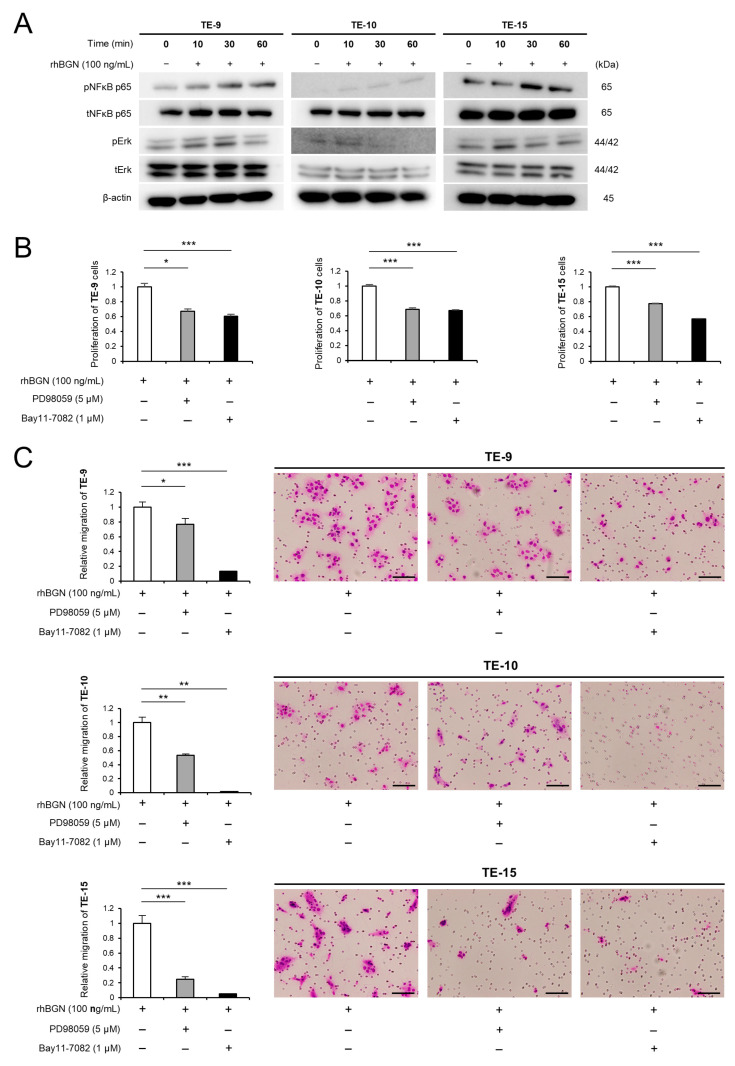
Biglycan (BGN) promotes the proliferation and migration of esophageal squamous cell carcinoma (ESCC) through the Erk and NF-κB pathways. (**A**) To investigate recombinant human BGN (rhBGN; 100 ng/mL)-induced signaling in ESCC cells, the protein expression levels of pNF-κB, tNF-κB, pErk, and tErk were examined by Western Blotting. ESCC cells were treated with rhBGN for 0, 10, 30, or 60 min. β-actin was used as a loading control. (**B**,**C**) MTS assays (**B**) and Transwell migration assays (**C**) were conducted to evaluate changes in the proliferation (**B**) and migration (**C**) of TE-9, TE-10, and TE-15 cells following treatment with rhBGN (100 ng/mL) in the presence of a NF-κB inhibitor (BAY11-7082; 1 μM/L) or a MEK1/2 inhibitor (PD98059; 5 μM/L). Migrated cells were counted in five representative microscopic fields after 48 h, and representative images are shown to the right of the graphs (**C**). Data are presented as the mean ± standard error of the mean (SEM) from three independent experiments (**B**,**C**). * *p* < 0.05, ** *p* < 0.01, *** *p* < 0.001. Scale bars: 100 μm (**C**).

**Figure 4 ijms-26-12024-f004:**
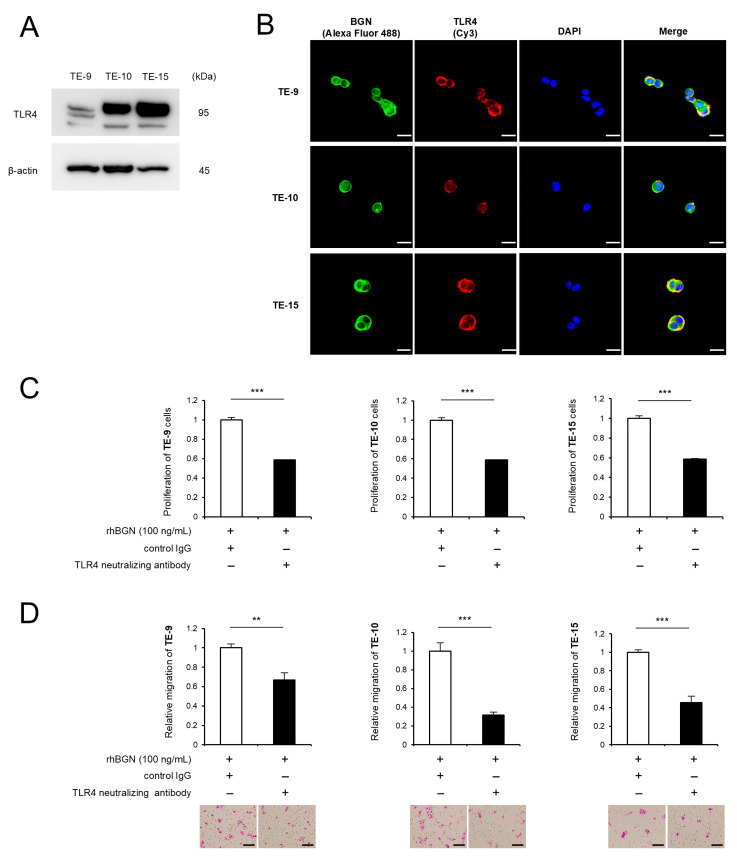
Biglycan (BGN) promotes proliferation and migration of esophageal squamous cell carcinoma (ESCC) cells through its receptor TLR4. (**A**) Expression of TLR4 in ESCC cells was confirmed by Western Blotting. (**B**) Double immunofluorescence staining for BGN (green) and TLR4 (red) was performed in ESCC cells treated with recombinant human BGN (rhBGN; 100 ng/mL). Nuclei were counterstained with DAPI (blue). (**C**,**D**) MTS assays (**C**) and Transwell migration assays (**D**) were conducted to evaluate changes in the proliferation (**C**) and migration (**D**) of TE-9, TE-10, and TE-15 cells following treatment with rhBGN (100 ng/mL) in the presence of a TLR4-neutralizing antibody (1 μg/mL) or control immunoglobulin G (IgG; 1 μg/mL). Migrated cells were counted in five representative microscopic fields after 48 h, and representative images are shown beneath the graphs (**D**). The data are presented as the mean ± standard error of the mean (SEM) from three independent experiments (**C**,**D**). ** *p* < 0.01, *** *p* < 0.001. Scale bars: 20 μm (**B**) and 100 μm (**D**).

**Figure 5 ijms-26-12024-f005:**
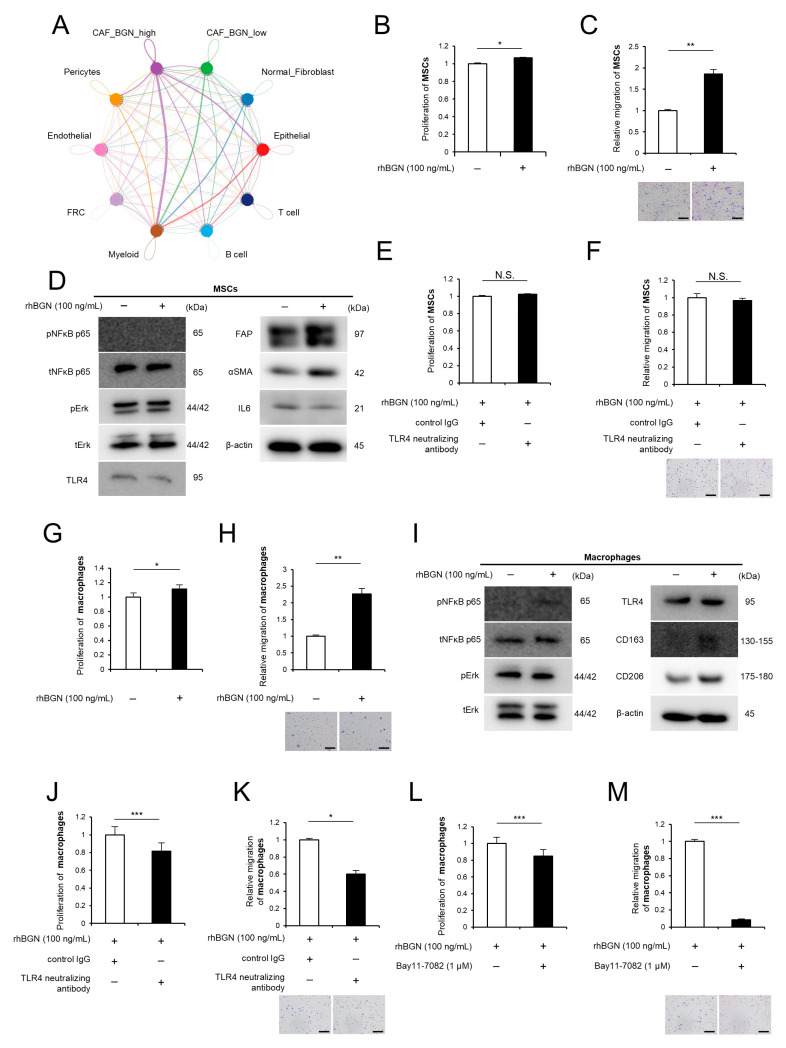
Biglycan (BGN) promotes proliferation, migration, and activation of macrophages and fibroblasts. (**A**) Cell–cell communication network analysis of *BGN*-expressing cancer-associated fibroblasts (CAFs) in esophageal squamous cell carcinoma (ESCC) tissues. Fibroblasts in single-cell RNA sequencing (scRNAseq) datasets from ESCC tissues were stratified into CAF_*BGN*_High (scaled expression > 3) and CAF_*BGN*_Low (≤3) groups based on *BGN* expression levels. Cell–cell communication inferred by CellChat revealed that CAF_*BGN*_High cells exhibited strong outgoing signaling toward epithelial and myeloid cells, as well as prominent autocrine signaling within the CAF_*BGN*_High population. (**B**,**C**) MTS assay (**B**) and Transwell migration assay (**C**) showing increased proliferation and migration of mesenchymal stem cells (MSCs) treated with recombinant human BGN (rhBGN; 100 ng/mL), respectively. (**D**) Western Blot analysis showing that fibroblast activation protein (FAP) and α-smooth muscle actin (αSMA) (CAF markers) were upregulated in MSCs treated with rhBGN for 24 h, whereas interleukin-6 (IL6) expression remained unchanged. TLR4 expression was detected; however, phosphorylation of NF-κB and Erk was not increased by rhBGN. (**E**,**F**) MTS assay (**E**) and Transwell migration assay (**F**) showing that rhBGN-induced proliferation and migration of MSCs were not suppressed by a TLR4-neutralizing antibody (1 μg/mL). (**G**,**H**) MTS assay (**G**) and Transwell migration assay (**H**) showing increased proliferation and migration of macrophages treated with rhBGN (100 ng/mL), respectively. (**I**) Western Blot analysis showing that rhBGN treatment upregulated the expression of CD163 and CD206 (M2 macrophage markers) and increased NF-κB phosphorylation, while TLR4 expression was also detected, but phosphorylated Erk was not changed. (**J**,**K**) MTS assay (**J**) and Transwell migration assay (**K**) showing that rhBGN-induced proliferation and migration of macrophages were attenuated by treatment with a TLR4-neutralizing antibody (1 μg/mL), respectively. (**L**,**M**) MTS assay (**L**) and Transwell migration assay (**M**) showing that treatment with the NF-κB pathway inhibitor Bay 11-7082 (1 μM) attenuated rhBGN-induced proliferation and migration to a similar extent, respectively. Data are presented as the mean ± standard error of the mean (SEM) from three independent experiments (**B**,**C**,**E**–**H**,**J**–**M**). * *p* < 0.05, ** *p* < 0.01, *** *p* < 0.001. N.S., not significant. Scale bars: 100 μm (**C**,**F**,**H**,**K**,**M**).

**Figure 6 ijms-26-12024-f006:**
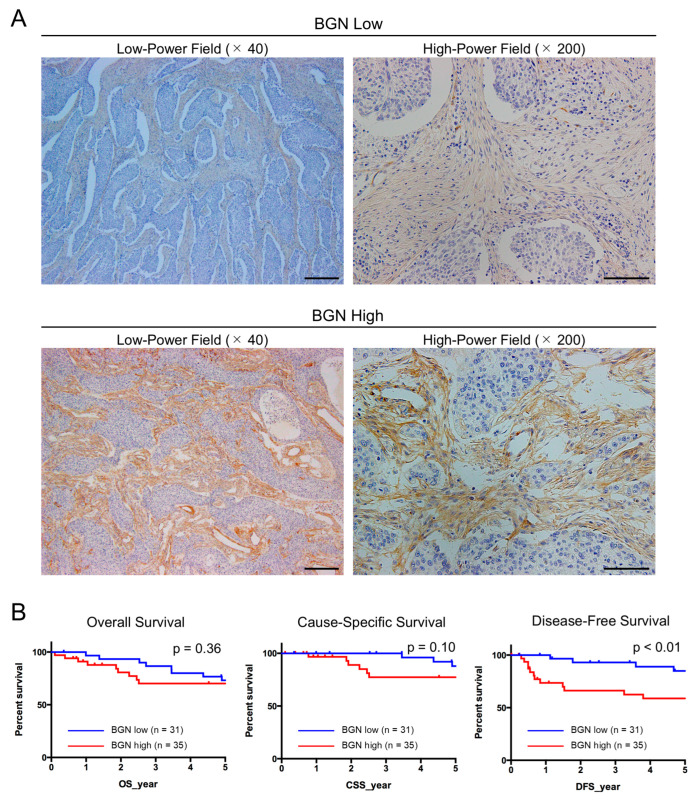
Biglycan (BGN) expression in cancer stroma of esophageal squamous cell carcinoma (ESCC) tissues and its association with clinical outcomes. (**A**) Immunohistochemical staining of BGN in ESCC tissues. Representative images are shown for low- and high-expression groups, stratified according to stromal BGN intensity at the invasive front (n = 31 and n = 35, respectively). (**B**) Kaplan–Meier curves showing overall survival, cause-specific survival, and disease-free survival, according to stromal BGN expression. Scale bars: 400 μm (**A**, left panels); 100 μm (**A**, right panels).

**Figure 7 ijms-26-12024-f007:**
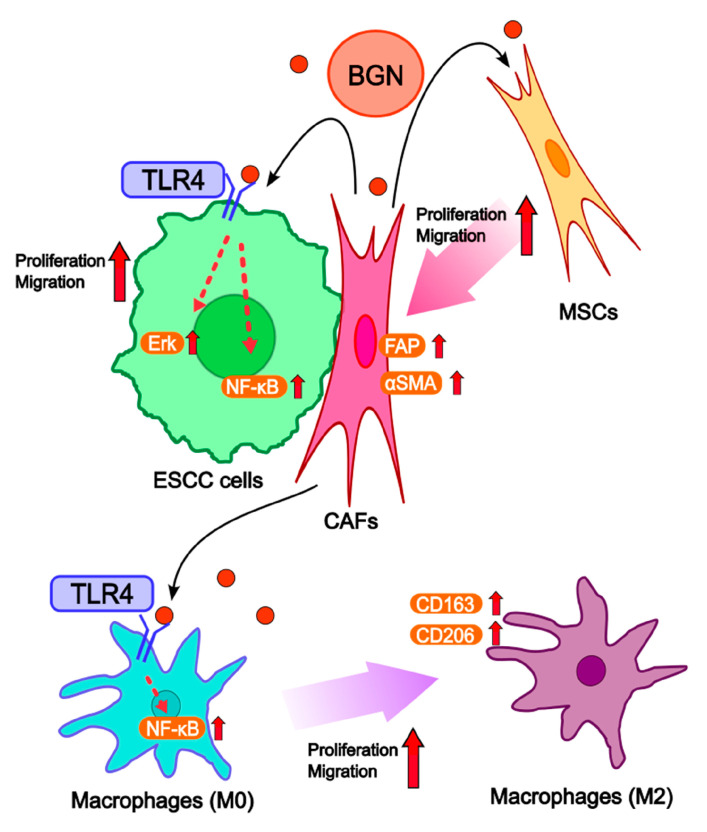
A schematic illustration of the role of BGN in the tumor–stromal interactions among ESCC cells, CAFs, and macrophages. BGN secreted from MSCs that have undergone CAF transition upon direct contact with ESCC cells promotes ESCC cell proliferation and migration through the TLR4–Erk/NF-κB signaling pathways. In addition, BGN enhances the proliferation and migration of MSCs and induces their differentiation into CAFs. Furthermore, BGN promotes the proliferation and migration of macrophages and drives their polarization toward the M2 phenotype through the TLR4–NF-κB signaling pathways.

**Table 1 ijms-26-12024-t001:** Genes markedly upregulated in CAF9 compared to MSC mono, but with low expression in TE-9 mono and TE-9 co.

Accession Number	Symbol	Description	Signal Intensity (Global Normalization)				Ratio	
			MSC mono	CAF9	TE-9 mono	TE-9 co	CAF9 /MSC mono	TE-9 co/TE-9 mono
NM_001711.5	*BGN*	Biglycan	2750	6702	40	76	2.44	1.92
NM_001164098.1	*VCAN*	Versican	1034	2355	45	62	2.28	1.37
NM_197955.2	*C15orf48*	chromosome 15 open reading frame 48	79	2346	36	52	29.88	1.48
NM_001856.3	*COL16A1*	collagen type XVI alpha 1	885	2139	30	50	2.42	1.67
XM_005271110.2	*P3H1*	prolyl 3-hydroxylase 1	494	1221	72	82	2.47	1.14
XM_005269005.2	*DRAM1*	DNA damage regulated autophagy modulator 1	375	1189	31	41	3.17	1.33
NM_000064.3	*C3*	complement component 3	28	1142	20	25	41.42	1.27
NM_001206.2	*KLF9*	Kruppel-like factor 9	358	1013	11	13	2.83	1.19

Signal intensity values correspond to cDNA microarray data (GSE244020 and GSE274064). MSC mono, mesenchymal stem cells cultured alone; CAF9, MSC-derived cancer-associated fibroblasts generated by direct co-culture with TE-9 cells; TE-9 mono, TE-9 cells cultured alone; TE-9 co, TE-9 cells cultured in direct co-culture with MSCs.

**Table 2 ijms-26-12024-t002:** Relationship of BGN Expression With Clinicopathological Features in ESCC Patients.

	Case Number	Expression of BGN		*p*-Value
		Low (n = 31)	High (n = 35)	
Age (years)				0.632
<65	32	16	16	
≥65	34	15	19	
Sex				0.798
Male	52	24	28	
Female	14	7	7	
Histological grade ^a^				0.575
SCCIS + WDSCC	15	8	7	
MDSCC + PDSCC	51	23	28	
Depth of tumor invasion ^a^				<0.001 *
T1	45	29	16	
T2 + T3	21	2	19	
Lymphatic vessel invasion ^a^				0.001 *
Negative	35	23	12	
Positive	31	8	23	
Blood vessel invasion ^a^				0.057
Negative	41	23	18	
Positive	25	8	17	
Lymphatic node metastasis ^a^				0.057
Negative	41	23	18	
Positive	25	8	17	
Stage ^b^				0.070
0 + I	37	21	16	
II + III + IV	29	10	19	
Expression of αSMA ^c^				0.003 *
Low	34	22	12	
High	32	9	23	
Expression of FAP ^c^				0.022 *
Low	37	22	15	
High	29	9	20	
Expression of CD68 ^c^				0.049 *
Low	32	19	13	
High	34	12	22	
Expression of CD163 ^c^				0.007 *
Low	31	20	11	
High	35	11	24	
Expression of CD204 ^c^				0.141
Low	32	18	14	
High	34	13	21	

Data were assessed using the chi-squared test. A *p*-value < 0.05 was considered statistically significant: * *p* < 0.05. ^a^ Classification was based on the 12th edition of the Japanese Classification of Esophageal Cancer [27,28]: SCCIS, squamous cell carcinoma in situ; WDSCC, well-differentiated squamous cell carcinoma; MDSCC, moderately differentiated squamous cell carcinoma; PDSCC, poorly differentiated squamous cell carcinoma. T1, tumor invades the mucosa and submucosa; T2, tumor invades the muscularis propria; T3, tumor invades the adventitia. ^b^ Tumor staging was based on the 9th edition of TNM classification by the Union for International Cancer Control [29]. ^c^ Patients were classified into low and high groups based on the immunoreactivity at the tumor-invasive front. The cutoff value for aSMA and FAP were set at 30% (high: above 30%, low: below 30%) [13].

## Data Availability

The data presented in this study are available on request from the corresponding author.

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
