# Peer review of "BGN Secreted by Cancer-Associated Fibroblasts Promotes Esophageal Squamous Cell Carcinoma Progression via Activation of TLR4-Mediated Erk and NF-κB Signaling Pathways"

_ijms, 2025, doi:10.3390/ijms262412024_

Round 1

Reviewer 1 Report

Comments and Suggestions for Authors

This study systematically explored the role of CAF-derived BGN in ESCC progression, finding that it activates the Erk and NF-κB signaling pathways through TLR4, promoting tumor cell proliferation and migration, and influencing macrophage polarization and CAF self-activation. The study design was relatively comprehensive, combining in vitro co-culture, single-cell transcriptomics, clinical sample analysis, and bioinformatics methods, demonstrating innovation and clinical significance. However, the article has some shortcomings in terms of mechanistic depth, experimental completeness, and data interpretation, and a major revision is recommended.

1. Currently, recombinant BGN and exogenous neutralizing antibodies are mainly used, lacking functional validation of endogenous BGN. It is recommended to use shRNA or siRNA to knock down BGN in CAF to verify its effect on ESCC cell proliferation and migration. Functional recovery experiments should be performed using conditioned media (CAF conditioned media vs. BGN knockdown conditioned media).

2. Although TLR4 is identified as a key receptor, the role of other known BGN receptors such as TLR2 has not been ruled out. It is recommended to observe whether BGN can still partially activate signaling pathways in TLR4 knockdown ESCC cells.

3. BGN promotes MSC proliferation, migration, and CAF marker expression, but does not activate NF-κB/Erk, and TLR4 antibody is ineffective, suggesting the presence of other receptors.

4. All mechanistic studies were conducted in vitro, lacking animal model support.

5. Although BGN was found to be highly expressed in both myCAF and iCAF, its role in CAF subtype transformation or functional heterogeneity was not further explored.

6. Quantitative analysis was not provided for some Western blot bands (e.g., p-Erk/p-NF-κB), and some images had low resolution.

Comments on the Quality of English Language

This study systematically explored the role of CAF-derived BGN in ESCC progression, finding that it activates the Erk and NF-κB signaling pathways through TLR4, promoting tumor cell proliferation and migration, and influencing macrophage polarization and CAF self-activation. The study design was relatively comprehensive, combining in vitro co-culture, single-cell transcriptomics, clinical sample analysis, and bioinformatics methods, demonstrating innovation and clinical significance. However, the article has some shortcomings in terms of mechanistic depth, experimental completeness, and data interpretation, and a major revision is recommended.

1. Currently, recombinant BGN and exogenous neutralizing antibodies are mainly used, lacking functional validation of endogenous BGN. It is recommended to use shRNA or siRNA to knock down BGN in CAF to verify its effect on ESCC cell proliferation and migration. Functional recovery experiments should be performed using conditioned media (CAF conditioned media vs. BGN knockdown conditioned media).

2. Although TLR4 is identified as a key receptor, the role of other known BGN receptors such as TLR2 has not been ruled out. It is recommended to observe whether BGN can still partially activate signaling pathways in TLR4 knockdown ESCC cells.

3. BGN promotes MSC proliferation, migration, and CAF marker expression, but does not activate NF-κB/Erk, and TLR4 antibody is ineffective, suggesting the presence of other receptors.

4. All mechanistic studies were conducted in vitro, lacking animal model support.

5. Although BGN was found to be highly expressed in both myCAF and iCAF, its role in CAF subtype transformation or functional heterogeneity was not further explored.

6. Quantitative analysis was not provided for some Western blot bands (e.g., p-Erk/p-NF-κB), and some images had low resolution.

Reviewer 2 Report

Comments and Suggestions for Authors

Cancer is a global health problem and a leading cause of death worldwide. According to the World Health Organization, 18 million people worldwide suffer from cancer, with 9.5 million cancer deaths recorded in 2018. According to the WHO, an estimated 20 million new cases of cancer and 9.7 million cancer deaths will be recorded in 2022. These grim statistics are currently changing unfavorably. Despite significant advances in the development of effective anticancer drugs, this field urgently needs new substances capable of acting on cellular targets or preventing the development of factors that promote tumor spread. In this study, the authors designed and conducted an investigation of the effects of biglycans, which have been shown to be generated by cancer-associated fibroblasts, on the development and migration of esophageal squamous cell carcinoma cells. As a result, BGN expression was shown to be significantly increased in CAFs in the ESCC tumor microenvironment and stimulate ESCC cell proliferation and migration by activating the Erk and NF-κB pathways via TLR4 signaling. Concurrently, BGN was shown to stimulate proliferation and migration in macrophages and induce M2-like polarization by activating the NF-κB pathway via TLR4, suggesting its multifaceted role in modulating the tumor microenvironment. These results support the potential of BGN as a therapeutic target and prognostic biomarker in ESCC. The article is clearly written, the experiment is very well designed, and it is clear that the article has been revised and the edits made for its improvement. However, it would be appreciated if the authors described possible ways and methods for mitigating the negative effects of both existing BGN and preventing its expression in the conclusion.

Reviewer 3 Report

Comments and Suggestions for Authors

This last revision of the article presents a well-structured and thorough research, supported by a carefully designed methodology. Authors have made all proposed clarifications and extensions in the text. It is clearly written and reflects meticulous execution throughout. The inclusion of recent and relevant references enhances the credibility of the study and indicates a comprehensive review of the existing literature. Overall, the paper is notable for its clarity, methodological rigor and depth. Figures, tables and figures have been carefully prepared and contribute substantially to the presentation and interpretation of the findings.This research has significant implications for a broad audience, including academic researchers, industry professionals, and diagnostic developers. The quality of the English language is strong and does not hinder comprehension. Formatting, structure, and tone are consistent and professional throughout, and no corrections or changes are required in the article.

I find the document satisfactory and ready to proceed for publishing.

Round 2

Reviewer 1 Report

Comments and Suggestions for Authors

The author has responded to the reviewers' concerns and revised the manuscript, I have no further comments.

Comments on the Quality of English Language

This study systematically explored the role of CAF-derived BGN in ESCC progression, finding that it activates the Erk and NF-κB signaling pathways through TLR4, promoting tumor cell proliferation and migration, and influencing macrophage polarization and CAF self-activation. The study design was relatively comprehensive, combining in vitro co-culture, single-cell transcriptomics, clinical sample analysis, and bioinformatics methods, demonstrating innovation and clinical significance. However, the article has some shortcomings in terms of mechanistic depth, experimental completeness, and data interpretation, and a major revision is recommended.

1. Currently, recombinant BGN and exogenous neutralizing antibodies are mainly used, lacking functional validation of endogenous BGN. It is recommended to use shRNA or siRNA to knock down BGN in CAF to verify its effect on ESCC cell proliferation and migration. Functional recovery experiments should be performed using conditioned media (CAF conditioned media vs. BGN knockdown conditioned media).

2. Although TLR4 is identified as a key receptor, the role of other known BGN receptors such as TLR2 has not been ruled out. It is recommended to observe whether BGN can still partially activate signaling pathways in TLR4 knockdown ESCC cells.

3. BGN promotes MSC proliferation, migration, and CAF marker expression, but does not activate NF-κB/Erk, and TLR4 antibody is ineffective, suggesting the presence of other receptors.

4. All mechanistic studies were conducted in vitro, lacking animal model support.

5. Although BGN was found to be highly expressed in both myCAF and iCAF, its role in CAF subtype transformation or functional heterogeneity was not further explored.

6. Quantitative analysis was not provided for some Western blot bands (e.g., p-Erk/p-NF-κB), and some images had low resolution.